# Effects of Commercial Probiotics on the Growth Performance, Intestinal Microbiota and Intestinal Histomorphology of Nile Tilapia (*Oreochromis niloticus*) Reared in Biofloc Technology (BFT)

**DOI:** 10.3390/biology13050299

**Published:** 2024-04-26

**Authors:** Ayesha Akter Asha, Mohammad Mahfujul Haque, Md. Kabir Hossain, Md. Mahmudul Hasan, Abul Bashar, Md. Zahid Hasan, Mobin Hossain Shohan, Nawshin Nayla Farin, Petra Schneider, Alif Layla Bablee

**Affiliations:** 1Department of Aquaculture, Bangladesh Agricultural University, Mymensingh 2202, Bangladesh; ayesha45310@bau.edu.bd (A.A.A.); mmhaque.aq@bau.edu.bd (M.M.H.); mahmudul.1506011@bau.edu.bd (M.M.H.); bashar43791@bau.edu.bd (A.B.); zahidhasan431139@gmail.com (M.Z.H.); shohan.1706097@bau.edu.bd (M.H.S.); farin.1906004@student.bau.edu.bd (N.N.F.); 2Department of Fisheries Management, Bangladesh Agricultural University, Mymensingh 2202, Bangladesh; kabir39284@bau.edu.bd; 3Department of Water, Environment, Civil Engineering and Safety, Magdeburg-Stendal University of Applied Sciences, 3655 Magdeburg, Germany; petra.schneider@h2.de

**Keywords:** *Oreochromis niloticus*, biofloc, probiotics, growth performance, histology, sustainable aquaculture

## Abstract

**Simple Summary:**

While various commercial probiotics are supplemented in biofloc technology (BFT), there is limited information regarding their impacts on farmed fish. This study investigated the effects of three commercial probiotics, two alone and one in combination with enzymes, on Nile tilapia reared in BFT. Incorporating multi-species probiotics along with enzymes into a BFT significantly enhanced tilapia weight gain, liver, and intestine weight, improving digestion and absorption responses, highlighting the potential benefits of BFT.

**Abstract:**

Though different types of commercial probiotics are supplemented in biofloc technology (BFT), very little information is available on their effects on the farmed fish. Therefore, this study focused on evaluating the effects of three most commonly used commercial probiotics on the growth performance, intestinal histomorphology, and intestinal microbiota of Nile tilapia (*Oreochromis niloticus*) reared in BFT. Tilapia fry, with an average weight of 3.02 ± 0.50 g, were stocked at a density of 60 fry/0.2 m^3^, and cultured for 90 days. Three commercial probiotics were administered, with three replications for each: a single-genus multi-species probiotic (*Bacillus* spp.) (T1), a multi-genus multi-species probiotic (*Bacillus* sp., *Lactobacillus* sp., *Nitrosomonas* sp., *Nitrobacter* sp.) (T2), and a multi-species probiotic (*Bacillus* spp.) combined with enzymes including amylase, protease, cellulase, and xylanase (T3). The results showed significant variations in growth and feed utilization, with T3 outperforming other treatments in terms of weight gain, liver weight, and intestine weight. Adding *Bacillus* spp. with enzymes (T3) to water significantly increased the histomorphological parameters (villi length, villi depth, crypt depth, muscle thickness, intestinal thickness) as well as microbes (total viable count and total lactic acid bacteria) of intestine of fish compared to T1 and T2, leading to improved digestion and absorption responses. It is concluded that the supplementation of commercial probiotics has potential benefits on farmed fish species in BFT.

## 1. Introduction

The escalating concerns regarding food security and nutrition, compounded by the challenges of population growth, climate change, and global warming, have prompted a growing focus on the potential of aquatic food production, notably through aquaculture [1,2]. Globally aquaculture production has experienced a rapid increase over the last five decades. Recent data indicate about 50% of the world’s fish supply is sourced from aquaculture [3,4]. Bangladesh, like other tropical countries, is widely recognized as one of the premier locations for freshwater aquaculture [5,6], due to its abundant resources and favorable agroclimatic conditions [7]. Bangladesh ranked fifth worldwide with a total fish production of 4.76 million MT in 2021–2022, while also holding the fourth position in tilapia production globally and the third in Asia [4,8]. Tilapia (*Oreochromis niloticus*) culture has gained popularity in developing countries [9], particularly among low-income countries like Bangladesh, for several compelling reasons. This fish is known for its rapid growth rate and resilience in various environmental conditions, making it well-suited for aquaculture systems in diverse settings, contributing to the livelihoods and food security of millions of resource-poor farmers [10,11]. Its ability to thrive in freshwater ponds, flood lands, and even in brackish water ponds further enhances its importance, offering flexibility to farmers with access to different farming practices [12,13,14]. 

While tilapia production holds great promise in aquaculture, its success is dependent upon access to costly feed resources and the maintenance of water quality parameters. In the context of Bangladesh, tilapia farming establishing a cost-effective feeding system is very challenging, because the feed cost ranges from 52–80% of the total production cost [15], since 45–50% of the fish feed ingredients in Bangladesh has to be imported [16]. To tackle these challenges, it is imperative to adopt new technological innovations and enhance traditional culture systems to optimize feed efficiency and lower production costs. Technological innovations are also crucial for fostering high-density tilapia farming within an effective feeding regime, utilizing basic natural resources like water, while minimizing the impact on the surrounding environment [17,18]. 

In any intensive aquaculture systems, increasing organic matter levels in the water caused by feces and leftover feed are alarming as fish can retain only about 20–30% of the feed nutrients given, while the rest 70–78% remains as faecal and metabolic waste in the water [19]. The breakdown of organic matter by microbes in the ammonification process can produce ammonia (NH_3_) in the waters. Feces and feed residues that accumulate in the water can increase the concentration of ammonia, which is toxic to the fish [20]. According to [21], ammonia in water causes fish to be susceptible to bacterial infections and have poor growth. Therefore, it is necessary to handle feed waste, feces, and water quality parameters in the culture systems. In this context, biofloc technology (BFT) emerges as a deliberate approach to aquaculture systems, where microbes play a crucial role in supporting fish growth by converting waste materials into protein-rich food, known as biofloc [22]. BFT promotes efficient resource utilization by converting excess nutrients and organic matter into microbial biomass, which serves as a natural food source for cultured species. This reduces the reliance on external feeds, thus lowering production costs and minimizing environmental pollution from uneaten feed and waste. BFT enhances water quality through the uptake of nitrogenous compounds, thereby reducing the need for water exchange and conserving precious freshwater resources, especially critical in Asian regions like Bangladesh where underground water is extensively utilized for aqua farming [23].

Probiotics are microorganisms utilized in the biofloc system that can play a crucial role in maintaining water quality by absorbing nitrogenous compounds. They produce in situ microbial protein, enhancing nutritional value by reducing feed conversion ratio, ultimately reducing feed costs and enhancing the health of cultured organisms [24,25]. The application of probiotics into ponds significantly alters the body compositions of fish, particularly in terms of crude protein, lipid content, moisture levels, and ash content [14,26]. In BFT, a specific carbon–nitrogen ratio (in general 10–15:1) induces the growth and proliferation of heterotrophic bacteria which efficiently reduce toxic nitrogenous compounds and keep water quality suitable for fish [27,28]. Therefore, biofloc, acting as a high-value supplementary feed, is naturally generated in controlled aquaculture systems, fostering the growth and production of fish [29,30,31]. 

The biofloc consists of various food organisms including bacteria, phytoplankton, yeast, rotifers, ciliates, protozoans, nematodes, copepods, and crustaceans [32,33,34,35]. These organisms can provide various nutrients such as complex carbohydrates from phytoplankton, proteins, and fatty acids for growth and production of fish. To digest those complex nutrient components effectively, fish requires various types of enzymes in their digestive system. The combination of various probiotics and enzymes can deliberately improve the digestion process in the digestive tract of fish [36]. Despite this potential, studies that integrate both probiotics and enzymes, and their application in the biofloc system remain limited. Combining these enzymes and probiotics could potentially yield a synergistic effect, providing a complementary approach to improving aquaculture sustainability [37,38]. Probiotics offer versatile strategies to enhance the productivity of aquaculture systems, including the utilization of a single bacterial genus, multiple genera, and the incorporation of enzymes along with either single or multiple bacterial genera [38]. 

Being a low trophic level fish in the food web, tilapia can efficiently consume biofloc and their amalgamated bacteria as a secondary food source [39]. BFT has been introduced recently in Bangladesh as well as in several other tropical countries with Nile tilapia (*O. niloticus*), climbing perch (*Anabas testudineus*), stinging catfish (*Heteropneustes fossilis*), and tiger shrimp (*Penaeus monodon*) [40]. Farmers are convinced by the field agents of aquaculture drugs and chemical companies and the shop-owners to apply commercial probiotics in BFT. Previously, some experimental studies were performed regarding the safety issues of biofloc-produced fish consumption [41], effects of stocking density of *P. monodon* in biofloc [42], and health risks of biofloc-produced fish [43]. However, the scientific information about the effects of commercial probiotics in development of biofloc as feed, and their extent of benefits alone or in combination with enzymes in growth performance as well as gut health of *O. niloticus* remains unrevealed. Therefore, this experimental study aimed to investigate the effects of supplementing three most commonly used commercial probiotics, two alone and one in combination with enzymes, on the growth performance, intestinal histomorphology, and intestinal microbiota of Nile tilapia (*O. niloticus*) reared in BFT. This study aims to identify which one is more feasible for biofloc aquaculture in Bangladesh.

## 2. Materials and Methods

### 2.1. Ethical Consideration

This experimental study was conducted based on the ethical guidelines developed by the Ethical Committee of Bangladesh Agricultural University Research System (BAURES), Bangladesh Agricultural University, Mymensingh, Bangladesh (977/BAURES/ESRC/FISH/31). 

### 2.2. Experimental Site and Design

This research was carried out at the Wet Laboratory of Faculty of Fisheries, BAU, Mymensingh for 90 days. Nine fiberglass tanks, each with a capacity of 200 L (≈0.2 m^3^) of water, were utilized, along with an aerator (Model: ACO-006) to ensure continuous air circulation within the water via air stones. Three most commonly used commercial probiotics (Pondcare, Aqualife, and Everfresh) were purchased from the manufacturing companies designated as treatment T1 (*Bacillus* spp.), T2 (*Bacillus* sp. + *Lactobacillus* sp. + *Nitrobacter* sp. + *Nitrosomonas* sp.) and T3 (*Bacillus* spp. + enzymes), with each treatment being replicated three times. The enzymes, including amylase (2000 IU), protease (3000 IU), cellulase (3800 IU), and xylanase (18,000 IU), were pre-mixed with the probiotic bacteria within the manufactured packet provided by the company, which was subsequently administered in T3. Prior to fish stocking, the experimental tanks underwent a thorough cleaning process with bleaching powder, followed by a two-day period of sun drying. 

### 2.3. Experimental Fish

Healthy fry of monosex Nile tilapia (*O. niloticus*) were collected from Reliance Aqua farm, Trishal, Mymensingh, Bangladesh which were transported in oxygenated polythene bags to the experimental site. Before stocking into triplicate tanks, tilapia fry were acclimatized by feeding on a basal diet comprising 37% crude protein, 8% crude lipid, 12% moisture, 16% ash, 4% crude fiber, 2.1% calcium, and 0.8% phosphorus. During the acclimatization period of 6 days, optimum aquatic environmental parameters such as a dissolved oxygen level of 10–11 mg/L, temperature range of 26–28 °C through water exchange, and NH_3_ levels below 0.05 mg/L were maintained. The fry were randomly allocated to nine aquaria measuring 3 ft × 2 ft × 1.5 ft and given 6 days to acclimate to the ambient temperature of the tanks. Subsequently, fry with an average weight of 3.02 ± 0.50 g were randomly placed into experimental tanks at a stocking density of 60 fry/0.2 m^3^.

### 2.4. Floc Preparation Using Commercial Probiotics

Before stocking fish, the floc was prepared applying the selected three commercial probiotics (Table 1), according to the method described by [19]. The recommended amount of probiotics was separately placed into three distinct plastic buckets and meticulously mixed with water to mitigate the risk of cross-contamination. To maintain C:N ratio of 15:1, 36.0 g of molasses, 36.0 g of limestone, 8.5 g of deionized salt, and 4.0 g of feed were added into each bucket containing 15.0 L water (Figure 1). To induce flock formation in the tank, continuous aeration was ensured to diffuse the air into the water via porous air stones. Every 15 days, one liter of liquid probiotics from each bucket was incorporated into the corresponding treatment tanks. The floc volume was measured fortnightly using Imhoff cones following the process recommended by [44]. The required amount of molasses was added daily to adjust the C:N ratio.

### 2.5. Feeding 

Commercial floating feed named Mega Fish Feed (containing 37% crude protein, 8% crude lipid, 12% moisture, 16% ash, 4% crude fiber, 2.1% calcium, and 0.8% phosphorus) was administered up to the satiation level of the experimental tilapia. Fish were fed 5% of total body weight on the first 30 days, and 3% of body weight on the second and third 30 days.

### 2.6. Water Quality Parameters

Water quality parameters underwent regular monitoring at four-day intervals throughout the duration of the experiment. Temperature and dissolved oxygen (DO) were measured through a portable DO meter (Model: Lutron DO-5509, Lutron, Coopersburg, PA, USA). A portable pH meter (Model: Hanna 981,017, Hanna, Smithfield, RI, USA), TDS meter (Model: TDS-3, HM Digital, Redondo Beach, CA, USA), and ammonia testing kits (API testing kit, Mars, Inc., McLean, VA, USA) were used to measure water pH, TDS, and ammonia, respectively.

### 2.7. Growth Monitoring and Data Recording 

Every 15 days, ten fish were randomly sampled from each tank for the purpose of measuring fish body weight gain, adjusting feeding amounts according to fish weight, and examining fish health. At the end of the experiment, the total count of individual fish in each tank, along with their respective weight and length measurements, were utilized to estimate both growth performance and survival rate. Fish weight was determined using a digital balance (Model: FSH, A&D Company Ltd., Seoul, Republic of Korea), while their length was measured using a measuring scale. 

The following formulae were used to calculate growth performance [45,46]: Wight gain = Final body weight − Initial body weight(1)
(2)$$Specific\;growth\;rate,SGR\left(\%/day\right)=\frac{\ln(Final\;weight)-\ln(Initial\;weight)}{Number\;of\;days\;reared}\times 100$$
(3)$$Survival\;rate,\left(\%\right)=\frac{Number\;of\;fish\;harvested}{Number\;of\;fish\;stocked}\times 100$$
(4)$$Feed\;conversion\;ratio,FCR=\frac{Dry\;feed\;fed\;(g)}{Live\;weight\;gain\;(g)}$$
(5)$$\mathrm{Viscerosomatic}\;\mathrm{index},\;\mathrm{VSI}\;(\%)=\frac{Visceral\;weight\;(g)}{Body\;weight\;(g)}\times 100$$
(6)$$\mathrm{Hepatosomatic}\;\mathrm{index},\;\mathrm{HSI}\;(\%)=\frac{Liver\;weight\;(g)}{Body\;weight\;(g)}\times 100$$
(7)$$\mathrm{Percent}\;\mathrm{weight}\;\mathrm{gain},\;\%\;\mathrm{WG}=\frac{Final\;weight\left(g\right)-Initial\;weight\;(g)}{Initial\;weight\;(g)}\times 100$$


### 2.8. Histology 

At the end of the feeding trial, nine tilapia from each treatment were randomly sampled and anesthetized with eugenol (90 ppm). Sterile scissors and forceps were used to collect the intestinal samples for histological study. The organs were then cleaned and stored in labelled bottles containing Bouin’s fluid (fixative agent) for 24 h. The fixed samples were moved to 70% alcohol for preservation and kept at 4 °C until they were ready for histological examination. Then, the fixed intestinal tissues were graded into an alcohol series, and molten wax was used to embed the samples. Trimming was performed on the prepared blocks, and the sections were cut at 10 μm thickness by a microtome machine. After staining the sections with hematoxylin–eosin, the intestinal morphological parameters were observed under a microscope (MCX100, Micros Austria, Sankt Veit an der Glan, Australia). Histological parameters were measured using image analysis application software Sigma Scan Pro5 (SPSS Inc., Chicago, IL, USA), as described by [47]. The histomorphological changes in the intestine tissues caused by the administration of probiotics were captured by a photomicroscope (AmScope 1000). 

### 2.9. Intestinal Microbiota Assessment

At the end of the experiment, entire gut microbiota specimens of six fish from each treatment were collected to determine the total viable count (TVC) and total lactic acid bacteria (TLAB) in the intestines of tilapia. The enumeration of TVC and TLAB was accomplished using the single plate–serial dilution spotting (SP–SDS) method [48]. Plate count agar was employed for determining TVC, while De Man, Rogosa, and Sharpe [MRS] agar was used for TLAB. Incubation procedures, following the guidelines of [49,50], were conducted separately to quantify the counts of TVC and TLAB. Then, the number of bacteria per milliliter (CFU/mL) was calculated by multiplying the number of colonies by the dilution factor.

### 2.10. Statistical Analysis

Data analysis was performed using the statistical package IBM SPSS Statistics 25.0. Prior to analysis, a normality test was performed to check the distribution of the data for normality. Subsequently, a homoscedasticity test was conducted, assuming equal or similar variances across the different treatment groups being compared. Data of all variables were run into one-way analysis of variance (ANOVA) and presented as mean ± standard deviation (SD). The significance of the difference between means was found out by Duncan Multiple Range Test at a 5% significant level (*p* < 0.05).

## 3. Results

### 3.1. Growth Performances of Fish

The mean final weight of tilapia (54.79 ± 5.03 g) was found significantly higher in T3 compared to other treatments (Table 2). Besides these, T3 showed the highest growth performance in different parameters such as weight gain (51.68 ± 4.83 g), intestine weight (2.27 ± 0.33 g), liver weight (1.65 ± 0.15 g) and these were significantly (*p* < 0.05) higher compared to fish of T1 and T2. In cases of weight gain (g) of fish, T3-multi-species probiotic + enzymes showed better response than T1 (42.11 ± 3.71) and T2 (43.54 ± 4.43). However, there is no significant (*p* > 0.05) effect of probiotic application on PWG (%), FCR, SGR (%/day), HSI, and VSI across all the treatments. The survival rates across the three treatments varied between 67.65% and 85.66%. However, a statistically significant difference (*p* < 0.05) was found, with T1 showing a lower survival rate (67.65 ± 10.93%) compared to T2 and T3.

### 3.2. Changes in Intestinal Microbiota

There was a significant increase in TVC in the intestine of the fish of T3 compared to the T1 and T2 (Table 3). Concurrently, there was a significant increase (*p* < 0.05) in the count of TLAB in the gut of fish from T3 compared to those in T1 and T2.

### 3.3. Intestinal Histomorphology and Digestive Response

Administration of probiotics significantly (*p* < 0.05) increased the length, width, and surface area of the villi, crypt depth, wall thickness, and muscle thickness in T3 compared to T1 and T2 (Table 4 and Figure 2. Among these, the thickness of the wall in T3 (13.46 ± 2.29 μm) increased more than two-fold than in T1 (6.77 ± 1.30 μm).

A significant (*p* < 0.05) increment of intestinal mucosal fold (25.3 ± 2.67 μm), width of lamina propria (11.5 ± 1.90 μm), width of enterocytes (6.9 ± 1.20 μm) and number of goblet cells (21.9 ± 3.14) were observed in T3 compared to T1 and T2 (Table 5 and Figure 3).

### 3.4. Water Quality Parameters

In this experiment, the water quality parameters such as pH, TDS, DO, NH_3_, and temperature did not significantly differ (*p* > 0.05) among the three treatments (Table 6). The water quality parameters were within the suitable range of fish rearing. However, ammonia and TDS exhibited comparably elevated levels in T2.

## 4. Discussion

Several studies suggested probiotics were commercially used in finfish and shellfish aquaculture in the substitution of antibiotics to accelerate growth, increase feed efficiency [51], and enhance fish production safety [52,53]. The growing body of literature indicates employing commercial probiotics in BFT improves the growth and production performance of fish [40]. This study demonstrates in the climatic conditions of Bangladesh, utilizing commercial probiotics in BFT leads to better growth performance in fish. The findings of this experimental investigation revealed application of probiotics in combination with enzymes (T3) significantly increased the WG (51.68 ± 4.83 g), as well as intestine weight (2.27 ± 0.33 g), and liver weight (1.65 ± 0.15 g) of experimental fish, in comparison to T1 and T2. The improved growth performance and feed efficiency of experimental fish experienced in this study is consistent with the outcomes of previous studies [38,54,55,56,57,58]. Ref. [55] performed an experiment with *O. niloticus* in BFT and found addition of probiotic significantly increased the growth performance of experimental fish. Reference [58] reported farmed tilapia in BFT showed better growth performance and immunity compared to control. The survival rate of tilapia in T1 within BFT system of this study was lower compared to the other treatment groups. The literature shows the survival rate of tilapia in a BFT exhibits a diverse range, influenced by various factors such as water quality, feeding, stocking density, and management practices [59]. The lower survival rate observed in T1 at the initial stage of the experiment may have been attributed to abrupt deterioration in water quality, potentially causing a delay in floc formation, and unknown reasons. The enhanced growth performance in treatment T3 in this study could be attributed to improved intestinal digestion, nutrient absorption, and the quality of water in the BFT system. 

The intestinal digestion and absorption of nutrients in experimental tilapia depended on the intestinal digestive enzymes as well as beneficial bacterial load and intestinal histomorphology. The synergistic combination of amylase, protease, cellulase, xylanase enzymes, and probiotics in treatment T3 demonstrated a significant positive impact on the digestion of feed/food organisms produced in biofloc and absorption of digested nutrients. Amylase and protease enzymes play crucial roles in breaking down carbohydrates (starch) and proteins, respectively, facilitating the efficient utilization of nutrients in the fish diet [60]. Cellulase and xylanase, on the other hand, contribute to the breakdown of complex plant materials like phytoplankton in probiotics, improving the digestibility of fibrous components [61]. When integrated with probiotics—beneficial microorganisms that promote a healthy microbial environment in the biofloc system—these enzymes enhance the overall digestive processes, leading to increased nutrient absorption and improved growth rates in fish. Since biofloc is a mixture of heterotrophic bacteria (mostly), crustaceans, copepods, nematodes, protozoans, ciliates, microalgae, yeast, and rotifers [31,32], they release exogenous enzymes [62] which act as the self-defense mechanism. Therefore, external supplemented enzymes may not be able to digest or deteriorate the biofloc being developed. A study showed additional probiotics can increase the utilization and assimilation of feed because microorganisms produce lipase, amylase, and protease enzymes so that the nutrients in the feed can be sufficiently exploited in digestion of fish [62]. Some experimental studies showed shrimps reared the biofloc system exhibited markedly better growth performance and enzyme functionalities [63,64].

Moreover, there was a significant (*p* <  0.05) increase in TVC and TLAB in the intestine of the fish of T3 compared to the T1 and T2. In aquatic systems, microbes play a key role in nutrient recycling, organic matter degradation, or serve as immune stimulators [65]. Therefore, the existence of microbes in the culture scheme demonstrates the health condition of the reared species. The application of probiotics with *Bacillus* spp. along with exogenous enzymes increases the microbial population in the gut of the host animal. Some studies showed existence of an enzyme-producing bacterial community in the gut of cultured species of aquatic organisms provides amenities, particularly improved digestion to the host species [66,67]. 

The primary function of the fish intestine is to digest and absorb food, while secondarily serving as a frontline barrier against infections [68]. This research showed administration of probiotics significantly (*p* < 0.05) increased the length, width, and surface area of the villi, crypt depth, wall thickness, and muscle thickness of the intestine of the experimental tilapia in treatment T3 compared to T1 and T2. Furthermore, a significant (*p* < 0.05) increment of intestinal mucosal fold (25.3 ± 2.67 μm), width of lamina propria (11.5 ± 1.90 μm), width of enterocytes (6.9 ± 1.20 μm), and number of goblet cells (21.9 ± 3.14) were observed in T3 in comparison to T1 and T2. The more the villus height increases, the larger the surface area of the intestine becomes, which aids in intestinal performance. A previous study showed increasing intestinal surface area has functionality on nutrient absorption, and it is proportionate to the villus height enhancement, which bolsters intestinal performance [69]. According to [70,71], enterocytes act as absorptive cells that regulate passage of nutrients. Boosting the number of goblet cells in the intestine accelerates the secretion of immunological substances as well as mucus [53,72], which directly or indirectly defends the fish against pathogens. According to [73], proper selection of potential probiotic microorganisms can exponentially extend intraepithelial lymphocyte and goblet cell of *O. niloticus*. It was revealed mucus produced from goblet cells aids in the trapping and removal of infections, fights harmful substances, acts as a protective barrier for the gastrointestinal tract and decreases dehydration [74]. Along with this, glycoproteins and mucopolysaccharides, two components of mucin, contribute to antagonism against pathogens and effectively contribute to improving the overall health of fish [75,76]. Lamina propria in the digestive tract harbors a large number of macrophages and neutrophils [77]. The thickness of lamina propria is related to the concentration of lymphoid cells. High concentrations of lymphoid cells have a major role in the immune response of fish [78]. Mucosal immunity is also demonstrated by fatty mucosal folds that affirm enlarged digestion, assimilation, and adsorption area in the digestive tract. Thus, exogenous probiotics and enzymes in the BFT system improved the digestion as well as absorption of nutrients and immunity of the experimental fish.

The experimental findings of this research showed all water quality parameters were within the acceptable limits for tilapia cultivation as formerly reported by [79]. Among the parameters, DO and ammonia play crucial roles in the physiology of fish. Unionized ammonia (NH_3_) and other nitrogenous waste are prime factors in intensive aquaculture operation, and according to [80], their mobility is important for preventing fish from sudden death and accelerating sustainability of the aquaculture system. The experimental findings showed water treated with probiotics could reduce ammonia range and median levels in rearing water. These results appeared to align with the findings from [81] that administration of two bacterial strains (*Pediococcus acidilactici* and *Bacillus cereus*) significantly decreased nitrogenous concentation in culture water. Additionally, [82] indicated adding probiotic strains (*B. amyloliquefaciens* and *B. cereus*) into tilapia (*O. mossambicus*) rearing water notably reduced ammonia concentration. The *Bacillus* species have the capacity to dispel different forms of nitrogen from host wastewater [83], a finding corroborated by [84]. This species plays a significant role in regulating the nitrogen cycle through processes such as ammonification, nitrification, denitrification, and nitrogen fixation. Thus, probiotics played a vital role in maintaining water quality parameters at optimum levels for fish. Under optimal conditions, fish experienced reduced stress and exhibited efficient feeding behavior, leading to enhanced growth.

The study is subjected to limitations stemming from the absence of a control group. Our aim was to examine variations in fish growth performance while maintaining the integrity of the biofloc system through the application of probiotics and enzymes, leading us to forgo the inclusion of a control group. Nevertheless, for a more comprehensive understanding, incorporating a single biofloc treatment as a control could enhance the study’s completeness, and this avenue can be explored in future research.

## 5. Conclusions

The ability of tilapia to thrive in various aquaculture setups underscores the significant necessity for implementing efficient production methods, particularly in developing countries like Bangladesh [85]. In this research, tilapia cultured in biofloc aquaculture system with the administration of commercial probiotics along with various enzymes resulted in significant improvements in growth and feed utilization through enhancing the gut microbial population, improvements in gut health, and morphology. From this experimental study, it is clear probiotics administered in combination with enzymes can be an important growth promoter for the Nile tilapia. As one of the most promising possibilities for producing food, BFT has attracted the attention of the scientific community and producers of sustainable aquaculture to ensure protein-rich food security for the growing population. However, the biofloc aquaculture system is not yet well-performing in Bangladesh. More studies are needed to determine which probiotic strains are best suited for the particular geographical areas of Bangladesh in the context of aquaculture practices. Although a novel technology may encounter initial challenges in adapting to a specific country’s context, continuous research efforts with nexus approach [86] can pave the way for successfully promoting and integrating the technology.

## Figures and Tables

**Figure 1 biology-13-00299-f001:**
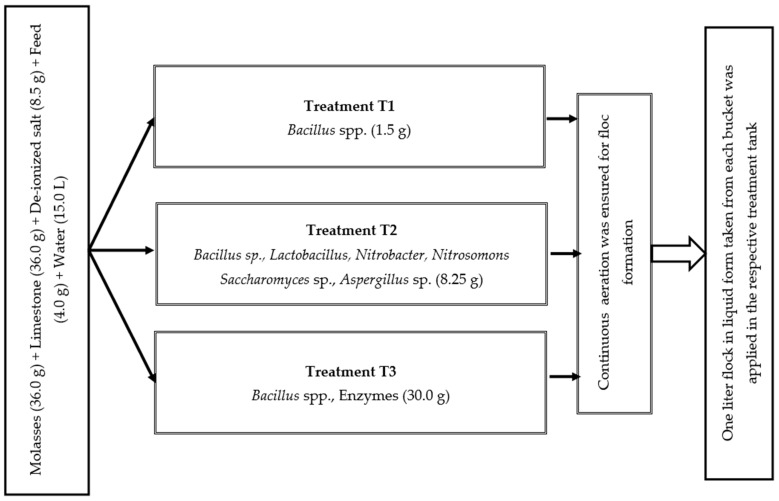
Floc preparation in three different treatments.

**Figure 2 biology-13-00299-f002:**
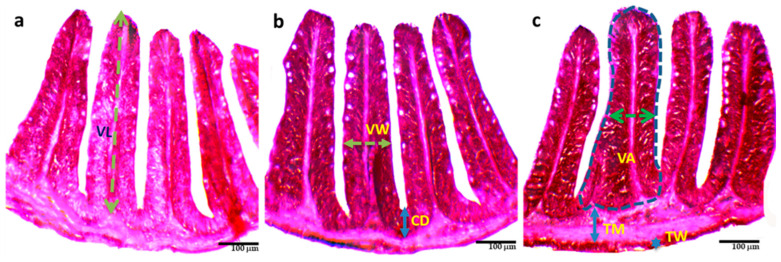
Histological changes (**a**–**c**) of the intestine of *O. niloticus* reared with different combinations of commercial probiotics, and enzymes in a biofloc system for 90 days; (**a**) = T_1_; (**b**) = T_2_; (**c**) = T_3_; TM = Thickness of muscular; TW = Thickness of wall; VW = Villus width; VL = Villus length; VA = Villus area; CD = Crypt depth.

**Figure 3 biology-13-00299-f003:**
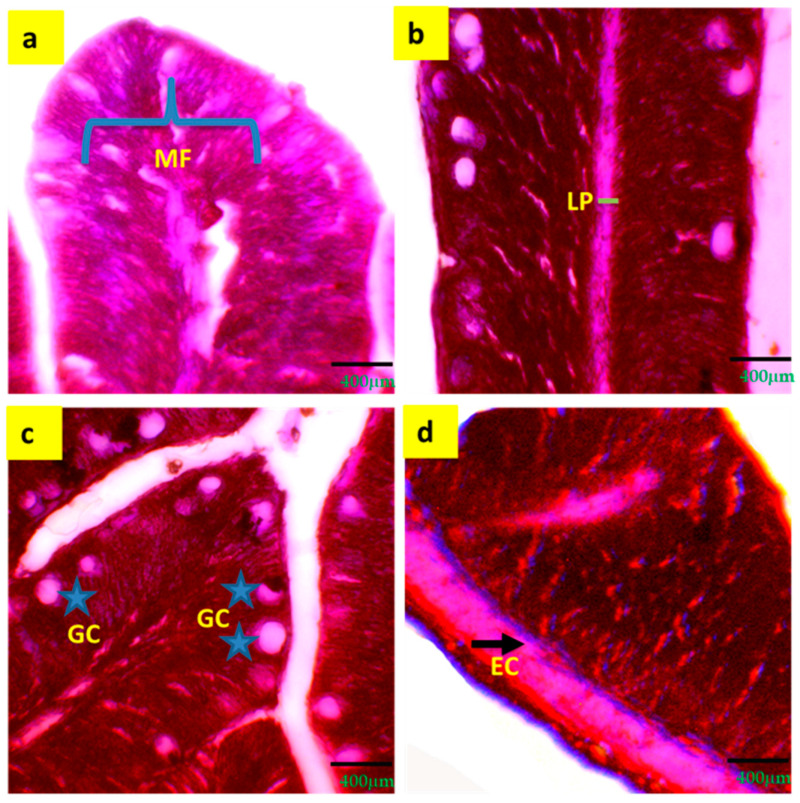
Digestive and immune response indicators (**a**–**d**) of histological gut of tilapia reared with a biofloc system for 90 days. (**a**) = MF (mucosal folds); (**b**) = LP (lamina propria); (**c**) = GC (goblet cell); (**d**) = EC (enterocyte). Arrows and grey stars indicate the specific locations of different intestinal parameters.

**Table 1 biology-13-00299-t001:** Composition and quantity (cfu/g) of probiotics.

Treatment 1 (Pondcare)	Treatment 2 (Aqualife)	Treatment 3 (Everfresh)
*Bacillus subtilis* (22 × 10^9^) *Bacillus licheniformis* (22 × 10^9^) *Bacillus polymyxa* (22 × 10^9^) *Bacillus pumilus* (22 × 10^9^) *Bacillus megaterium* (22 × 10^9^) *Bacillus coagulans* (22 × 10^9^) *Bacillus amyloliquefaciens* (22 × 10^9^)	*Bacillus subtilis* (10 × 10^9^) *Bacillus licheniformis* (7 × 10^9^) *Bacillus mensentericus* (9 × 10^9^) *Lactobacillus acidophilus* (9.8 × 10^9^) *Nitrobacter* sp. (7.5 × 10^9^) *Nitrosomonas* sp. (8 × 10^9^)	*Bacillus subtilis* (5 × 10^9^) *Bacillus licheniformis* (5 × 10^9^) *Bacillus megaterium* (2.5 × 10^9^) *Bacillus pumilis* (2.5 × 10^9^) Amylase (2000 IU) Protease (3000 IU) Cellulase (3800 IU) Xylanase (18,000 IU)

**Table 2 biology-13-00299-t002:** The effects of different commercial probiotics on the growth performance, feed utilization, and survival rate of tilapia (*O. niloticus*) under the biofloc system.

Growth Parameters	T1	T2	T3	*p*-Value	Level of Significance
IFL (cm)	4.53 ± 0.50	4.53 ± 0.50	4.53 ± 0.50	1.000	NS
FFL (cm)	13.15 ± 0.72	13.17 ± 0.53	12.95 ± 0.58	0.446	NS
IFW (g)	3.09 ± 0.55	3.11 ± 0.57	3.11 ± 0.57	0.989	NS
FFW (g)	45.20 ± 3.77 ^a^	46.66 ± 4.61 ^a^	54.79 ± 5.03 ^b^	0.000	**
Weight gain (g)	42.11 ± 3.71 ^a^	43.54 ± 4.43 ^a^	51.68 ± 4.83 ^b^	0.000	**
PWG (%)	1395.48 ± 234.31	1431.40 ± 253.75	1536.92 ± 320.96	0.241	NS
Intestine weight (g)	1.63 ± 0.39 ^a^	1.83 ± 0.15 ^b^	2.27 ± 0.33 ^c^	0.000	**
Liver weight (g)	1.14 ± 0.36 ^a^	1.24 ± 0.34 ^a^	1.65 ± 0.15 ^b^	0.000	**
FCR	0.73 ± 0.07	0.71 ± 0.07	0.67 ± 0.10	0.052	NS
SGR (%/day)	1.12 ± 0.08	1.12 ± 0.08	1.15 ± 0.10	0.401	NS
HSI	2.53 ± 0.83	2.67 ± 0.73	3.03 ± 0.40	0.068	NS
VSI	3.62 ± 0.86	3.95 ± 0.49	4.19 ± 0.81	0.061	NS
Survival rate (%)	67.65 ± 10.93 ^a^	85.99 ± 10.14 ^b^	85.66 ± 1.57 ^b^	0.000	**

Values with different superscript letters in a row differ significantly (*p* < 0.05). The tabled values are presented as mean ± SD; NS = not significant; ** = significant at 1% significance level. IFL = Initial fish length (cm); FFL = Final fish length (cm); IFW =Initial fish weight (g); FFW = Final fish weight(g); PWG = Percent weight gain (%); FCR = Feed conversion ratio; SGR = Specific growth rate (% per day); HSI = Hepatosomatic index; and VSI = Viscerosomatic index.

**Table 3 biology-13-00299-t003:** TVC and total LAB in the intestines of tilapia at the end of the experiment.

Gut Microbial Content	Treatments			*p*-Value
	T1	T2	T3	
TVC (×10^8^ CFU/mL)	2.6 ± 0.45 ^a^	2.8 ± 0.12 ^a^	4.63 ± 1.62 ^b^	0.000
TLAB (×10^4^ CFU/mL)	0.67 ± 0.13 ^a^	0.83 ± 0.23 ^a^	1.33 ± 0.18 ^b^	0.000

Values with different superscript letters in a row differ significantly (*p* < 0.05) among the treatments. All values expressed as mean ± SD, (*n* = 9); TVC = Total viable count (CFU/mL); TLAB = Total lactic acid bacteria (CFU/mL).

**Table 4 biology-13-00299-t004:** Gut morphology of Nile tilapia in different treatments.

Gut Morphological Parameters	T1	T2	T3	*p*-Value	Level of Significance
Villi length (μm)	332.84 ± 22.94 ^a^	335.69 ± 9.62 ^a^	443 ± 40.75 ^b^	0.000	**
Villi width (μm)	64.69 ± 7.39 ^a^	62.69 ± 5.58 ^a^	70.69 ± 8.45 ^b^	0.021	*
Villi area (μm^2^)	21,517.07 ± 2816.32 ^a^	21,062.61 ± 2163.40 ^a^	31,305.69 ± 4675.42 ^b^	0.000	**
Crypt depth (μm)	26.15 ± 2.88 ^a^	29.08 ± 8.06 ^a^	36.61 ± 3.94 ^b^	0.000	**
Wall thickness (μm)	6.77 ± 1.30 ^a^	8.46 ± 2.18 ^b^	13.46 ± 2.29 ^c^	0.000	**
Muscle thickness (μm)	27.38 ± 8.22 ^a^	29.15 ± 4.23 ^a^	37.38 ± 3.77 ^b^	0.000	**

Values with different superscript letters in a row differ significantly (*p* < 0.05). All Values are expressed as mean ± SD. * = significant at 5% level; ** = significant at 1% level.

**Table 5 biology-13-00299-t005:** Gut histological parameters reflecting both digestive and immune responses in tilapia.

Gut Histological Parameters	T1	T2	T3	*p*-Value	Level of Significance
Fattening of mucosal folds (μm)	19.8 ± 4.44 ^a^	18.6 ± 2.88 ^a^	25.3 ± 2.67 ^b^	0.000	**
Width of lamina propria (μm)	5.6 ± 2.01 ^a^	5 ± 1.63 ^a^	11.5 ± 1.90 ^b^	0.000	**
Enterocyte width (μm)	3.9 ± 1.10 ^a^	3.5 ± 0.71 ^a^	6.9 ± 1.20 ^b^	0.000	**
Abundance of goblet cell (GB)	13.4 ± 5.62 ^a^	13.3 ± 4.16 ^a^	21.9 ± 3.14 ^b^	0.000	**

Values with different superscript letters in a row differ significantly (*p* < 0.05). Values are presented as mean ± SD. ** = significant at 1% significance level.

**Table 6 biology-13-00299-t006:** Water quality parameters in experimental tanks for rearing tilapia.

Parameters	T1	T2	T3	*p*-Value
pH	8.30 ± 0.67	8.28 ± 0.71	8.29 ± 0.63	0.995
TDS (mg/L)	553.35 ± 233.19	643.65 ± 323.37	533.60 ± 191.10	0.354
DO (mg/L)	7.81 ± 2.55	7.64 ± 2.50	7.21 ± 2.54	0.746
Ammonia (ppm)	0.19 ± 0.37	0.21 ± 0.41	0.18 ± 0.36	0.951
Temperature (°C)	27.25 ± 1.77	27.4 ± 1.85	27.35 ± 1.76	0.964

Values are presented as mean ± SD. TDS = Total dissolved solids; DO = Dissolved oxygen.

## Data Availability

The supportive data of the findings of this study are available from the corresponding author upon request.

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
