# Peer review of "Effects of Commercial Probiotics on the Growth Performance, Intestinal Microbiota and Intestinal Histomorphology of Nile Tilapia (Oreochromis niloticus) Reared in Biofloc Technology (BFT)"

_biology, 2024, doi:10.3390/biology13050299_

Round 1

Reviewer 1 Report

Comments and Suggestions for Authors

Biofloc technology is known to be a modern method in aquaculture to reduce production costs and minimise the impact on the environment. In the current study, the authors investigated the effects of three mostly used commercial probiotics on the growth performance, intestinal histomorphology and intestinal microbiota of Nile tilapia reared in biofloc technology. It was shown that multispecies probiotic (Bacillus spp.) combined with enzymes (amylase, protease, cellulase, xylanase) improved growth, feed utilization, the histomorphological parameters and intestinal microbiota. Obtained results are interesting. However, there are some comments that should be addressed to enrich the quality of the manuscript.

1.      Three types of probiotics were used - a single-genus multi-species probiotic (Bacillus spp.), a multi-genus multispecies probiotic (Bacillus sp., Lactobacillus sp., Nitrosomonas sp., Nitrobacter sp.), and a multispecies probiotic (Bacillus spp.) combined with enzymes. Why wasn’t the combination (Bacillus sp., Lactobacillus sp., Nitrosomonas sp., Nitrobacter sp. + enzymes) used in the trial?

2.      Abstract. Line 28. Please add information about which enzymes were used.

3.      Materials and methods. Please provide information on the commercial names of the probiotics, feeds, enzymes used in the experiment.

4.      Please specify which protease was used and with what type of activity.

5.      Lines 183-185. Please specify which parameters were assessed every 15 days. How the catch of fish every 15 days has helped to estimate the survival rates? Why are the data in Table 2 only given for the start and the end of the experiment if fish parameters were estimated every 15 days?

6.      Line 61, 63. Remove the surnames from the quadruple brackets, leave just the numbers.

Author Response

Many thanks to the anonymous reviewer for the valuable instructions to make the revised manuscript more accurate and robust. All the suggested changes and revisions have been addressed in the revised manuscript. Responses to the comments of the reviewer are listed below. 

Reviewer 2 Report

Comments and Suggestions for Authors

This paper (Biology 2949623) entitled "Effects of Commercial Probiotics on the Growth Performance, Intestinal Histomorphology and Intestinal Microbiota of Nile Tilapia (Oreochromis niloticus) Reared in Biofloc Technology (BFT)" mainly compared the detailed effects of three commonly commercial probiotics alone and one in combination with enzymes on growth, intestinal morphology, and gut microbiota in Nile Tilapia cultured in BFT system. The addition of probiotics with enzymes exhibited superior performance in growth parameters and intestinal immune function than the other two treatments. Results from this study could highlight the possibility and feasibility of probiotics alone or in combination with enzyme to improve growth and health of Nile Tilapia in BFT system and provide the necessary foundation for developing safe, effective, and sustainable probiotics in aquaculture.

The core content of this manuscript is valuable for farmed aquatic animalsHowever, the manuscript contains confusing and redundant presentation in the main text. In this case, the authors need to proofread throughout this paper to increase its readability before it is accepted.

Major comments:

1. In the section of "1.Introduction", several sentences were lengthy, redundant, and confusing. For example, in Line 88-91, it contains long description that seems to be confusing to the reader. In Line 125-128, it contains the sentence that could be deleted without influencing the meaning of the whole paragraph. The authors may reorganize the relevant statement and streamline the presentation of known findings for laying out more clearly background in this study.

2. In this study, samples were obtained from Nile Tilapia treated with three different probiotics. Was there a blank control group without probiotics treatment? When Nile Tilapia in BFT did not treat with probiotics, were there any differences in the relevant parameters evaluated in this study? The author should give an explanation or provide more supporting data in the revised paper.

3. Prior to the feeding experiments, aquatic animals are cultured for one or two weeks to adapt to the laboratory conditions. What about the aquatic environmental parameters under acclimatization conditions? These parameters were same as those in the feeding experiment or not? Please provide more information about it in the section of "2.3. Experimental Fish".

4. Based on the statement in Line 153, basal diet was used in Nile tilapia during the acclimating period. So, the formulation and proximate composition of this basal diet should be included in this paper. The authors should provide the relevant information to "2.3. Experimental Fish" section.

5. The representation of the original "Discussion" section might be confusing to the reader. For example, in Line 327-329, please rephrase this part for a clarity of exposition. The discussion contents of this paper should be discussed and written with emphasis for better clarifying your findings in this study.

6.The authors should check the references format carefully after reading the Instructions for Authors, especially the name of co-authors in the references should be cited all or cited the first ten authors, note the cited journals should be abbreviated according to ISO 4 rules, DOI, the publication year, the web site of online version.

Minor comments:

1. The reference number should be placed in the main text by using square brackets "[ ]", rather than the author name and publication year. For example, Line 61-62, please remove ".Ali et al., 2023". Also, there were similar errors in the other part of this paper. Please check and correct accordingly.

2. In Fig.1, what was the addition level of T2 used in the experimental diet? 1.5 g, 30.0 g, or the other added amount? Please check and modify it in the revised figure.

3. The authors should check what is suggested for symbol formatting of volume unit according to the information to related guides. For example, Line 212, please replace "CFU/ml" with "CFU/mL".

4. In "3.Results" section, there were some redundant descriptions of results. The contents of this section should be simplified. For example, it would be preferable to directly delete the statement in Line 243-244. Please check the whole text and revise accordingly. 

5Table 4 lacks the interpretation of the letter superscript in a row. It is suggested that the authors provide more information in the legend of Table 4 for more clearly showing results. There was a similar problem in Table 5.

6. Figure 3 lacks the sub-captions or legends of Fig.3a-3d. Do they represent the intestinal staining results in T1, T2, and T3? The authors should provide all complete and correct legend of Figure 3 in the revised version, or explain this.

Other errors (highlighted in yellow) were marked in the PDF file.

Therefore, this manuscript will be reconsidered after major revision.

Comments on the Quality of English Language

This manuscript (Biology 2949623) entitled "Effects of Commercial Probiotics on the Growth Performance, Intestinal Histomorphology and Intestinal Microbiota of Nile Tilapia (Oreochromis niloticus) Reared in Biofloc Technology (BFT)" is meaningful for aquaculture and aquafeed. But some descriptions in this paper were confusing and a little bit redundant, particularly the section of "Introduction" and "Discussion". Also there are still several mistakes, such as the presenting symbols for volume unit. It is recommended that the text should be proofread by a professional or native speaker.

Author Response

We are grateful to the anonymous reviewer for the critical observation throughout the manuscript to make it more robust and suitable for publication. The suggested changes and revisions have been adopted in the revised manuscript. Responses to the comments of the reviewer are listed below.

Reviewer 3 Report

Comments and Suggestions for Authors

1. Fish density in fish/m3

2. Line 162-163: avoid using terms such as  "certain amount". Describe in detail the quantities and the protocol used to form the bioflocs and reference it. I believe that if it is well explained in the methodology, figure 1 can be discarded.

3. How was aeration maintained? Blower? Did it reach the tank through hoses? Porous stones? Describe the experimental design better

4. Line 166- 167: put the reference used for this process

5. Standardize the font throughout the text, especially after line 188 where the formulas for calculating the zootechnical performance are found.

Author Response

Thanks to the reviewer for providing the suggestions. We have addressed all the changes and revisions as suggested (yellow shaded text). Thanks.

Round 2

Reviewer 2 Report

Comments and Suggestions for Authors

This resubmitted paper (Biology 2949623) entitled "Effects of Commercial Probiotics on the Growth Performance, Intestinal Histomorphology and Intestinal Microbiota of Nile Tilapia (Oreochromis niloticus) Reared in Biofloc Technology (BFT)" has been adequately modified in response to the comments of the reviewers. The authors have addressed the comments/reasons for the unchanged section in the revised manuscript. But the format of reference number in the main text should be carefully checked. In my view, the current manuscript should be accepted after minor revision.

Comments on the Quality of English Language

This resubmitted paper (Biology 2949623) entitled "Effects of Commercial Probiotics on the Growth Performance, Intestinal Histomorphology and Intestinal Microbiota of Nile Tilapia (Oreochromis niloticus) Reared in Biofloc Technology (BFT)" has been adequately modified in response to the comments of the reviewers. The authors have addressed the comments/reasons for the unchanged section in the revised manuscript. But the format of reference number in the main text should be carefully checked. In my view, the current manuscript should be accepted after minor revision.